# Anti-Inflammatory Effect of Izalpinin Derived from *Chromolaena leivensis*: λ-Carrageenan-Induced Paw Edema and In Silico Model

**DOI:** 10.3390/molecules28093722

**Published:** 2023-04-26

**Authors:** Juan C. Mancipe, Pedro Vargas-Pinto, Oscar E. Rodríguez, Paola Borrego-Muñoz, Iovana Castellanos Londoño, David Ramírez, Luis G. Piñeros, María Camila Mejía, Luis M. Pombo

**Affiliations:** 1Facultad de Ciencias Agropecuarias, Universidad de la Salle, Bogotá 110141, Colombia; jcmancipe@unisalle.edu.co (J.C.M.);; 2Facultad de Ingeniería, Universidad del Bosque, Bogotá 110121, Colombiapaola.borrego@juanncorpas.edu.co (P.B.-M.); luis.pineros@juanncorpas.edu.co (L.G.P.);; 3Escuela de Medicina, Fundación Universitaria Juan N. Corpas, Bogotá 110311, Colombia; 4Departamento de Farmacología, Facultad de Ciencias Biológicas, Universidad de Concepción, Concepción 4030000, Chile

**Keywords:** Anti-inflammatory, *Chromolaena leivensis*, flavonoids, izalpinin, creatinine kinase, molecular docking, molecular dynamics

## Abstract

The flavonoid izalpinin was isolated from the aerial parts of *Chromolaena leivensis*. Its structural determination was carried out using MS and NMR spectroscopic techniques (^1^H, ^13^C). This compound was evaluated for its anti-inflammatory effect in a rat model on λ-carrageenan-induced plantar edema. Paw inflammation was measured at one-hour intervals for seven hours following the administration of λ-carrageenan. Serum creatine kinase (CK) levels were evaluated, obtaining statistically significant results with the treatments at doses of 10 mg/kg (* *p* < 0.01) and 20 mg/kg (** *p* < 0.005). The anti-inflammatory effect of the compound was evaluated by using plethysmography, and the results showed significant differences at the three concentrations (10 mg/kg, 20 mg/kg, 40 mg/kg) in the first and third hours after treatment. * *p* < 0.05; ** *p* < 0.001; **** *p* < 0.0001 vs. the negative control group treated with vehicle (DMSO). Lastly, molecular docking analyses reveal that izalpinin has a strong binding affinity with five target proteins involved in the inflammatory process. The analysis using molecular dynamics allowed demonstrating that the ligand–protein complexes present acceptable stability, with RMSD values within the allowed range.

## 1. Introduction

Inflammation is a localized nonspecific immune response that manifests itself in response to any type of injury, causing increased blood flow, vasodilation, fluid extravasation, increased cell metabolism, and mediator response [1,2]; additionally, on physical examination, there is evidence of redness, pain, heat, swelling, and loss of function. Inflammation is a common and important pathological process in medicine [3]. However, without proper treatment, it can lead to the development of chronic diseases, such as rheumatoid arthritis, periodontal disease, cardiovascular disease, cancer, and Alzheimer’s disease [4,5,6].

The search for active phytopharmacological principles could contribute to finding molecules as an alternative for the management of inflammation and to minimize the side effects of traditional treatments [7]. In this field, flavonoids have emerged as potential candidates since, in recent years, it has been reported that they possess anti-inflammatory properties involving the inhibition of the activity of several pro-inflammatory biochemical mediators, for instance, cytokines, chemokines, adhesion molecules, prostanoids, NO•, enzymes (COX-2, LOX, and iNOS), signaling pathways, and transcription factors [8,9]. Several studies have reported that these secondary metabolites have the ability to enhance endogenous anti-inflammatory chemical mediators (IL-10) and various antioxidant and detoxification defensive enzymes [10].

Among the enzymes that mediate the inflammatory process is cyclooxygenase (COX, prostaglandin H2 synthase (PGHS)), the key enzyme in the biosynthesis of prostaglandin from arachidonic acid. It is bifunctional since it catalyzes both the conversion of arachidonic acid to prostaglandin G2 (PGG2) and to prostaglandin H2 (PGH2) as peroxidase. COX exists in two isotypes: COX-1 and COX-2 [11]. COX-2 is an inducible enzyme that participates in the induction of a response under pro-inflammatory conditions [12].

Among the species that have reported the presence of flavonoids is *Chromolaena leivensis* (Hieron.), an endemic plant from Colombia. This species has been used in traditional medicine as an analgesic, antitumor, and cold remedy [13]. Chemical composition studies have shown that the main chemical components of this species are flavones, such as 3,5-dihydroxy-7-methoxyflavone or izalpinin (Figure 1) [14]. However, there are no reports of anti-inflammatory activity for this species; hence, it is important to study this species and provide information about new natural products with anti-inflammatory activity.

In an attempt to design, develop, and understand the mechanism of action of therapeutic molecules, computational techniques have emerged as tools to minimize the time and resources necessary for their chemical synthesis and in vitro and in vivo assays [12]. As part of these computational techniques, we found molecular docking, a key method used in the design of computer-assisted drugs to predict molecular interactions at close distances. Its objective is to predict binding positions between a small molecule and a macromolecular target, which are referred to as a ligand and a receptor, respectively [15]. In addition to evaluating ligand orientation and receptor complex stability, it is an important factor in the process of computer-assisted drug design that can be validated by means of molecular dynamics [16].

In this study, flavone izalpinin (IZP) was isolated and identified by evaluating its anti-inflammatory activity in an animal model of acute inflammation using λ-carrageenan in Wistar rats, determining creatine kinase levels, and by an in silico analysis using molecular docking and molecular dynamics in order to understand its mechanism of action on receptors involved in the inflammatory process.

## 2. Results and Discussion

### 2.1. Structural Determination of Izalpinin

Izalpinin (IZP) was obtained as a yellow solid with a melting point of 198 °C, soluble in acetone, and with a positive Shinoda test, which suggests that it is a flavonoid. In the GC–MS analysis, the chromatogram showed a majority signal at 8.49 min, and in its mass spectrum, a molecular ion of 284 [M^+^] (Appendix A). The ^1^H NMR spectrum (Table 1) showed a characteristic singlet of a phenolic OH group [δ_H_ 12.3 (s, OH)], two signals from aromatic protons [δ_H_ 6.77 (d, *J* = 1.8 Hz, 1H), 6.37 (d, *J* = 1.8 Hz, 1H)], a characteristic signal of methoxy hydrogens [δ_H_ 3.86 (s, 3H)] corresponding to ring A. Aromatic proton signals δ_H_ 8.19 (d, *J* = 7 Hz, 2H) and 7.61–7.47 (m, 3H) correspond to ring B (Appendix A). The ^13^C NMR spectrum presented 14 carbon signals (Table 1) attributable to a signal corresponding to a C-4 carbonyl group [δ_C_ 176.4], 7 quaternary aromatic C atoms, which include at least 5 oxygenated-C signals at δ_C_ 165.1, 160.4, 156.3, 146.1, and 137.4. The signals at δ_C_ 127.5 and 128.5 are equivalent to the carbons at the C-2′, C-6′, C-3′, and C-7′ positions, respectively, of ring B. The signals at δ_C_ 160.4 and 137.4 confirm the presence of the C–OH at C-5 and C-3 (Appendix A).

### 2.2. Anti-Inflammatory Activity of Izalpinin

λ-carrageenan can elicit an acute inflammatory response with the infiltration of polymorphonuclear neutrophils and increased exudate in pleurisy [17]. Figure 2 shows the anti-inflammatory effect of IZP at different times of experimentation. There was significantly less plantar edema in the treatments and the positive control, demonstrated by the percentages of inflammation and the histopathological evaluation, showing an anti-inflammatory effect of IZP by decreasing vasodilation and vascular permeability. The greatest effect occurred after the third hour, where the second phase of acute inflammation begins. In this model, at this point, there was an increase in the production of COX-2 and its derivatives, such as prostaglandins, which could indicate that the main effect of the flavone results from these enzymes and their products [18].

CK levels were obtained from whole blood samples at the end of the study and evaluated using chemiluminescence (Figure 3). In low-dose treatments (10 and 20 mg/kg), values are lower compared to the positive and negative controls, showing an effect on this indicator of muscle damage due to inflammation, with statistical significance * *p* < 0.01; ** *p* < 0.005 [19].

Histological sections (Figure 4) also evidenced less infiltration of polymorphonuclear cells in the treated groups, which developed mild (b) and moderate (c) dermatitis; however, in the group treated with DMSO 1%, there were cases of suppurative dermatitis (d), with greater infiltration of inflammatory cells in subcutaneous tissue and muscle, in addition to increased edema. Similarly, there is greater preservation of the structure of the muscle fibers in the treated groups. Taking into account cell infiltration and the extent of tissue damage, this may be due to the lower number of inflammatory mediators and cy-tokines that influence vasodilation, proliferation, and chemotaxis of inflammatory cells [20]. Figure 4 shows sections of control group (healthy animals) (a), and treatments with IZP at a dose of 20 mg/kg (b), diclofenac positive control (c), and negative control (Vehicle-DMSO 1%).

IZP has shown anti-inflammatory effects due to reducing edema and serum CK levels, which indicates less damage to skeletal muscle and infiltration of polymorphonuclear cells in plantar tissue, as well as the preservation of its histological structure. More research studies are necessary to find the mechanism and site of action in the inflammation cascade and to continue advancing in the evaluation of this flavone with anti-inflammatory potential, so that it can be used jointly in the treatment of inflammatory conditions. The PMNs present in five fields were counted using a 100x objective. Only differences (lower number of PMNs) are observed in the group treated with IZP at 20 mg/kg (Appendix A).

### 2.3. Binding Interaction of Izalpinin with Key Targets

Molecular docking studies followed by binding free energy calculations were conducted to study the binding mechanism of IZP with the enzymes involved in the inflammatory process. The results obtained by docking and MM-GBSA of IZP are shown in Table 2.

Figure 5 showed the interactions among enzymes implicated in the inflammatory process (COX-2, HAase, 5-LOX, NOS, and 5-PLA2) and IZP. The interactions between IZP and COX-2 (PDB code: 5KIR) observed in the 2D and 3D diagrams show that IZP has mostly hydrophobic interactions with residues Y348, V349, L352, Y355, F381, L384, Y385, W387, A516, I517, F518, M522, V523, and A527. Polar interactions were observed with H90, G192, S353, and S530. Residues R120 and R513 showed electrostatic interactions. H90 also showed a hydrogen bond with the methoxy group at position C7. T355 exhibited this type of interaction with the hydroxyl group at position C5, while S530 did so with the hydroxyl group at position C3. Residues Y385 and W387 exhibited a π–π stacking interaction with flavonoid ring B (Figure 5a). Ferreira et al. found that the residues R120 and Y355 make up the entrance to the active site, while Y385 and Y348 are at the apex of the active site [21].

With the HAase enzyme (PDB code: 2PE4), IZP showed polar interactions with N37, N39, and S323. Hydrophobic interactions were observed with residues P62, V322, and W324 with the flavonoid’s ring B, while residues I73, Y75, V127, Y202, Y247, Y286, W321, and V322 with rings A and C of IZP. Similarly, it was observed that residues Y75 and W321 presented π–π stacking interactions with ring C, while Y75 presented this interaction with ring A. W324 presented a π–π stacking interaction with ring B. Electrostatic interactions were observed with residues D129 and E131. Hydrogen bonding interactions were observed between the oxygen of the pyran ring (ring C) and residue W21, while residue E131 exhibited this same interaction with the hydroxyl group of ring A position 5 (Figure 5b). The most important interactions between IZP and HAase are mainly hydrophobic, electrostatic, and hydrogen bonds. Our results are in line with those obtained by Li et al. (2021), where they were able to show that the binding site of the six flavonoids evaluated is located near residues Y75, T202, Y247, W321, and W324, which indicates that the most important interaction forces between these flavonoids and HAase are mainly hydrophobic and hydrogen bonds [5]. With regard to HAase, an important role in catalysis is suggested for E131 and a supporting role for D129 [22]. In addition, Y202 and Y247 are important residues for catalytic activity, given that Y202 possibly binds to the substrate, and Y247 apparently coordinates and stabilizes oxidation during the formation of the transition state [23].

Hydrophobic-type interactions with residues Y81, F169, A398, F402, I406, N554, Q555, F610, L615, A672, and P668 were observed with the 5-LOX receptor (PDB code: 3V99). Residues Q15, N669, and S670 exhibited polar-type interactions, while residues R401, E614, and E622 exhibited electrostatic interactions. Hydrogen bonding interactions were observed between R401 and the hydroxyl group at position 5, between E614 and the hydroxyl group at position 3, as well as between residue S670 and the carbonyl group of ring C. Residue F169 exhibits an interaction with ring C, while F402 does so with ring A (Figure 5c). Lipoxygenases are enzymes related to the oxidation of various fatty acids and Ras proteins. IZP showed several interactions with the residues of the active site N554, G555, and the nearby lipophilic amino acid A672. In case of 5-LOX, the pocket is relatively large and flexible [24].

IZP with the NOS receptor (PDB code: 2NSI) showed hydrophobic interactions with residues M120, I201, M374, W461, I462, W463, V465, P466, P467, and F476. Residues E377 and R381 showed electrostatic interactions with ring B. The hydroxyl group of position 3 exhibited a hydrogen bond interaction with the HEME group. T376 exhibited a polar interaction (Figure 5d). The results obtained from molecular docking made it possible to show that IZP presented interactions with residues of the active site W461, E377, I462, and W463 [25].

PLA2-IIA has a well-preserved region, which is composed by a hydrophobic site where the fatty acid tails of the substrates interact, and a catalytic site for substrate cleavage. The hydrophobic site contains aliphatic and aromatic residues within or near the N-terminal helix (L2, F5, I9, A17, A18, Y21, C28, C44, and F98). The “catalytic site” is formed by the hydrophilic residues H47 and D48, which together with the catalytic Ca^2+^ are necessary for the enzyme cutting mechanism consisting of the nucleophilic attack of a water molecule on the diacylglycerol sn-2 acyl ester bond, given that PLA2 substrates interact with catalytic Ca^2+^ [26]. In the case of PLA2-IZP, hydrophobic interactions were observed with L2, F5, I9, A17, A18, Y21, C28, V30, C44, Y51, and F98. H47 exhibited a π–π stacking interaction with rings A and C, while Y51 exhibited a π–π stacking interaction with ring A, and F5 with ring C. D48 exhibited a hydrogen bonding interaction with the 5-position hydroxyl group, of the electrostatic type. Polar-type interactions with H6 and H47 are also present (Figure 5e).

### 2.4. Molecular Dynamics Studies of IZP Interacting with Key Enzymes COX-2, HAase, 5-LOX, NOS, and 5-PLA2 Involved in Inflammation

The five complexes were subjected to molecular dynamics simulations (MDs). The stability of each target and ligand in all complexes was evaluated by measuring the root-mean square deviation (RMSD) of the target backbone and IZP atoms (Figure 6). All five targets remained stable during the 100ns-MDs; no major changes were detected in backbone atoms. The time-dependent distance between the different positions of the set of atoms by the root mean square fluctuation (RMSF) was also evaluated to measure the IZP behavior through the simulation (Appendix A).

IZP remains stable when interacting with COX-2 (Figure 6a). The RMSF value was 1.5 Å for atoms 16, 18, 20, which corresponds to B-ring atoms (Appendix A). Interaction analysis (Appendix A) showed hydrophobic interactions with residues V349, L352, Y355, Y385, W387, A516, F518, V523, and A527. H-bonds were observed with residues Y355 and R527. In the interaction scheme (Appendix A), electrostatic interactions were observed with R120 with the carbonyl groups at C3 and hydroxyl at the C5 position and R513 with the methoxyl group at the C7 position. During MD simulations, an interaction with residue R513 was observed that was not detected in molecular docking. This interaction is believed to play a crucial role in differentiating the active sites of isoforms structurally by affecting the ligands behavior (Appendix A) [12]. This is why many compounds such as izalpinin can selectively inhibit COX-2 [21].

Figure 6b shows that HAase does not change significantly; this is due to the restriction applied to the atoms of the secondary structure and to the stability of the crystal during the simulation. IZP showed a significant change after 40 ns, which corresponds to the free rotation of the flavonoid B-ring bond; after this time, the conformation remains stable. The RMSF value was 4.5 Å, and the atoms that showed major changes were 12, 16 to 20 corresponding to the B-ring (Appendix A). In the interaction analysis, hydrophobic interactions were observed with residue P62, while with residues Y202, Y286, and W321, hydrophobic and water bridge-mediated hydrogen bond type interactions were observed. Y75 and W324 exhibited hydrophobic and water bridge-mediated interactions (Appendix A). The interaction scheme (Appendix A) showed π–π stacking interactions with hydrophobic residues such as Y75, Y202, Y275, and W321 with the A and C rings of IZP.

The 5-LOX protein did not show significant RMSD changes during the simulation, while IZP showed a significant change around 50 ns explained by conformational changes due to free rotation of O-CH_3_ in the position C7 and flavonoid B-ring bond. These changes led to rapid stabilization at the binding site (Figure 6c); the RMSF value was around 0.8 Å, and atoms 2 (O-CH_3_), 16, 18, and 20 (B-ring) showed these changes (Appendix A). In the analysis of interactions, hydrophobic-type ones were observed with residues F402, F555, and P668. F610 and A672 show hydrophobic and water bridge-mediated interactions. E614, L615, and N669 exhibited hydrogen bond type and water bridge-mediated interactions (Appendix A). In the interaction scheme (Appendix A), electrostatic interactions were observed with E614 and the hydroxyl group at C5 and N669 with the carbonyl group at C3, as evidenced in MM-GBSA.

For the NOS–IZP interaction, the ligand showed an RMSD of 3Å, which may indicate that it is undergoing conformational changes within the active site due to solvent exposure allowing some bonds (O-CH_3_, ring B) to rotate freely (Figure 6d). The methoxy group atoms at C7 (2 and 21) and the atoms present in ring A (13, 14, 15) presented an RMSF value of approximately 1Å, due to the free rotation of these bonds. (Appendix A). Hydrophobic and water bridge-mediated interactions with residues R381, W463, and P467 were observed in the graph. Residues M374 and W461 showed hydrophobic interactions, while I462 had hydrogen-bond, hydrophobic, and water bridge-mediated interactions (Appendix A). In Appendix A, π-cation interaction with R381 and the A and C rings of IZP was observed, while W463 presented π–π stacking interaction with C ring.

In the 5-PLA2–IZP complex, IZP showed a shift around 70 ns with a RMSD value of 2 Å possibly due to conformational change of free rotation of bonds O-CH_3_ and ring B, in order to stabilize quickly and fit properly into the binding site. The RMSF value was 1.5 Å (Appendix A) for atoms 2 (O-CH_3_), 11, 13, 14, and 15 (ring A). Interaction analysis revealed that residues F5, A17, and Y51 exhibited hydrophobic interactions. H27 showed interactions mediated by water bridges. Residue H47 exhibited hydrogen bond, hydrophobic, and water bridge-mediated interactions (Appendix A).

## 3. Materials and Methods

### 3.1. General Information

Melting points (MP) were measured on a MELTEMP fusiometer (Laboratory Devices. Mass 02139, Cole-Parmer Ltd., Staffordshire, UK). The measured temperatures were expressed in °C. Gas spectrometry combined with mass spectrometry (GC–MS) was obtained by coupling an AGILENT TECHNOLOGIES GC 6850 II equipment to an AGILENT TECHNOLOGIES MS 5975B mass spectrometer. A temperature program was used starting at 70 °C, maintaining it for 2 min; then a ramp of 20 °C/min was applied until reaching 270 °C, maintaining it for 4 min; the injection volume was 1μL in split mode with a flow of 1 mL/min and a split ratio of 10. A mass spectrometer was used, with the form of electron impact ionization at 70 eV; the temperature of the transfer line was 250 °C, and a simple quadrupole was used as an analyzer. The ^1^H and ^13^C NMR (nuclear magnetic resonance, Bruker, Billerica, MA, USA) spectra were taken on a BRUKER AMX-300 spectrometer (300 MHz for ^1^H, 100 MHz for ^13^C), using DMSO as solvent and tetramethylsilane (TMS) as internal standard at room temperature. Chemical shifts are reported in δ (ppm) regarding TMS, while J-coupling constants are reported in Hertz. Additionally, compound purification was carried out using flash column chromatography using silica gel 60 (230–400 mesh). The purity of the compound was verified by using thin-layer chromatography performed on aluminum plates coated with 0.2 mm thick silica gel F_254_ (Merck, Darmstadt, Germany). The compound was visualized by adsorption under UV light (254 nm–365 nm). 

### 3.2. Plant Material

The plant material was collected on the outskirts of the city of Bogotá, Cundinamarca via Chusacá; 4°33′0.14″ N; 74°15′0.41″ W (complete plant in flowering stage). A control sample was sent to the National Herbarium of Colombia for its taxonomic identification, where it was classified as *Chromolaena leivensis* (Hieron). King & H. Rob (Bogotá, Colombia), with voucher number COL 535219.

### 3.3. Extraction and Isolation

The leaves of *C. leivensis* (660 g) were extracted with ethanol (95%) using Soxhlet equipment (Sovirel, Bogotá, Colombia). The solvent was removed under reduced pressure, obtaining an ethanolic extract (280 g). An amount of 200 g of the ethanolic extract were taken to perform a column fractionation using petroleum ether (5.3 g), toluene (18.1 g), dichloromethane (21.3 g), and ethyl acetate (19.3 g) as solvents. Three grams of the toluene fraction were taken and carried to a column chromatography with silica gel using a stepped gradient of petroleum ether: toluene (8:2, 3:1, 1:1), toluene, toluene: methanol (95:5), and methanol to obtain 52 fractions that were monitored using TLC. Fractions 6–8 were joined by chromatographic similarity to obtain the flavonoid izalpinin (70.4 mg). Yellow solid; pf 197–198 °C; ^1^H NMR (300 MHz, DMSO) y ^13^C NMR (100 MHz, DMSO) spectroscopic data for characterization of compound are presented in Table 1.

### 3.4. Anti-inflammatory Assay

#### 3.4.1. Experimental Animals

The procedure was approved by the Research Committee of the Fundación Universitaria Juan N. Corpas (serial ID1964 code INV-P-44). Male Wistar rats between 8 and 10 weeks of age and weighing between 180 and 210 g were used. The animals were randomly organized into groups with six individuals each, arranged in standard boxes at room temperature (20 ± 2 °C) and 12-h light/dark cycles, and were given a balanced diet (LabDiet) and water ad libitum. The animals were treated in accordance with ethical criteria and animal experimentation protocols, pursuant to OECD Guideline 404 [27].

#### 3.4.2. Evaluation of the in Vivo Anti-Inflammatory in Rat Paw Edema Induced by λ-Carrageenan

Izalpinin (IZP) isolated from *C. leinvensis* was resuspended in 1% DMSO, and the anti-inflammatory activity was determined through the rat paw edema assay using three doses: 10 mg/kg, 20 mg/kg, and 40 mg/kg. The experimental subjects were organized into six groups (Table 3). Treatments were administered intraperitoneally (i.p.) one hour before 1% λ-carrageenan in the paw pad [28]. Edema-induced limb enlargement was measured by immersion with plethysmography (Ugo Basile Plethysmometer, model 7140, Germonio, Italy) at 1, 3, 5, and 7 h after the injection of the mucopolysaccharide. Measurements were made in triplicate [29].

#### 3.4.3. Creatine Kinase (CK) Assay and Histopathological Study

At the end of the study, whole blood samples were obtained from the lateral tail vein and preserved in a tube without anticoagulant to measure the serum creatine kinase (CK) value using chemiluminescence [30]. The samples from the paw pads were processed in accordance with the protocols reported by Zhang et al. (2013), by staining with hematoxylin and eosin, and were evaluated with a conventional optical microscope (Carl Zeiss, Göttingen, Germany) at 10×, 40×, and 100× magnification [31]. Cell counting was performed using a Zeiss optical light microscope (Carl Zeiss, Göttingen, Germany). The PMNs present in five fields were counted using a 100× objective. The counting was performed by two observers, and the results were averaged.

#### 3.4.4. Statistical Analysis

The statistical analysis for inflammation percentage was performed using Dunnett’s two-way ANOVA and post hoc; for the CK analysis, a one-way ANOVA (Kruskal–Wallis test; Dunn’s post hoc) was carried out using the GraphPad Prism software version 9.5.1 for Windows (GraphPad Prism version 9.5.1 for Windows, GraphPad Software, San Diego, CA, USA, www.graphpad.com). The threshold for significance was at *p* < 0.05.

### 3.5. In Silico Study

#### 3.5.1. Protein Selection Preparation and Grid Generation

The crystal structures of the selected proteins were retrieved from the protein data bank (PDB data base, www.rcsb.org (accessed on 20 September 2022)). The Protein Preparation Wizard module of the Schrödinger suite was used to prepare the proteins. The structures were processed by removing water molecules, adding hydrogen atoms, and filling the chains and loops. Models refinement was performed by energy minimizing the structures employing the OPLS3 force field [32].

#### 3.5.2. Structural Optimization of IZP

The ligand IZP were sketched with the Maestro Suite and prepared with the LigPrep module (Schrödinger, LLC, New York, NY, USA, 2020). Ligand energy minimization was performed using the OPLS3 force field [33].

#### 3.5.3. Molecular Docking and Binding Free Energy Calculation

Grid boxes were centered at reported binding sites for each target, outer box edges were set to 10Å, thus resulting in a box volume of 10 × 10 × 10 Å^3^ for all the enzymes. Molecular docking was performed with the Glide software (Schrödinger, LLC, New York, NY, USA, 2020), using the standard precision (SP) mode, that uses hierarchical filters to find the best ligand poses in the protein binding site. The filters incorporate a systematic search of orientation space, positional and conformational, of the ligands before assessing the energetic interaction with the protein [34]. All docking were re-scored by binding free energy calculations using the MM-GBSA (molecular mechanics combined with the generalized Born surface area) method. The idea is to estimate more accurate binding free energies for ligand–protein complex and to get a more reliable ranking. The implementation of MM-GBSA was carried out with the Prime module from Schrodinger [32]. The PLIP server [35] were used to visualize interactions of ligand–protein complexes, and visualization was carried out with Pymol [36].

#### 3.5.4. Molecular Dynamics Simulations (MDs)

MDs were carried out in Desmond [37]. Systems were solvated with the simple point charge (SCP) water model in an orthorhombic box. Na^+^ or Cl^-^ ions were added to neutralize the systems, and the final concentration of NaCl was set at 0.15 M. Then, each system were equilibrated for 20 ns using an NPT ensemble at 300 K with the application of a restraint spring constant of 5 kcal·mol^−1^·Å^−2^ to the protein backbone atoms. After a proper system equilibration, a 100 ns production MDs in the NPT ensemble was performed applying a restraint spring constant to 1 kcal·mol^−1^·Å^−2^ to the secondary structure. In both equilibrium and production MDs, temperature and pressure were kept constant at 300 K and 1.01325 bar, respectively, by coupling to a Nose–Hoover [38] thermostat method with a relaxation time of 1 ps and Martyna–Tobias–Klein barostat [39] to keep the pressure constant with an integration time of 2 ps. Trajectories were monitored using geometric properties throughout the simulation time; the most common measure of these deviations or structural fluctuations over time is made by calculating the root-mean-square deviation (RMSD) and root mean square fluctuation (RMSF) [34]. The simulations were analyzed with Desmond and VMD [40].

## 4. Conclusions

Izalpinin has shown anti-inflammatory effects due to reducing edema and serum CK levels, which indicates less damage to skeletal muscle and the infiltration of polymorphonuclear cells in plantar tissue, as well as the preservation of its histological structure. More research studies are necessary to find the mechanism and site of action in the inflammation cascade, and to continue advancing in the evaluation of this flavone with anti-inflammatory potential so that it can be used jointly in the treatment of inflammatory conditions. Lastly, molecular docking analyses show strong binding affinities between izalpinin and the proteins involved in the inflammatory process. Molecular docking analysis was validated by the results of 100 ns molecular dynamics simulations. From this analysis, ligand–protein complexes were shown to have good stability with RMSD values within the allowed range. Therefore, these experimental and theoretical results data will be useful in this study in understanding the mechanism of inhibitory effects of IZP against proteins involved in inflammation and explain the anti-inflammatory mechanism of flavonoid as anti-inflammatory drugs.

## Figures and Tables

**Figure 1 molecules-28-03722-f001:**
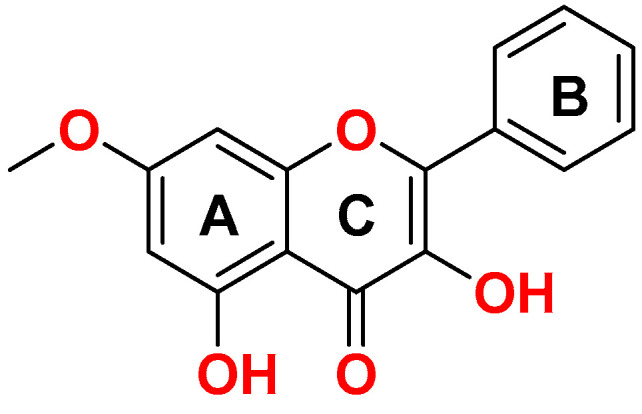
Structure of IZP.

**Figure 2 molecules-28-03722-f002:**
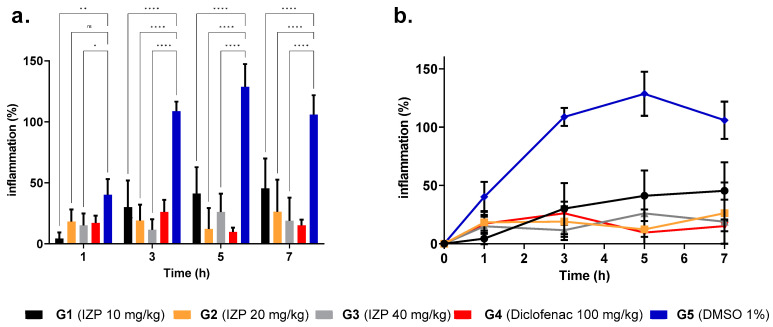
Anti-inflammatory effect of izalpinin on λ-carrageenan-induced edema in rat paw. (**a**) G5 negative control, treated with DMSO 1% (blue bar), has statistically significant differences compared to the treated groups (G1 10 mg/kg, G2 20 mg/kg, G3 40 mg/kg, and G4 diclofenac 100 mg/kg) in the first and third hours after treatment. * *p* < 0.05; ** *p* < 0.001; **** *p* < 0.0001; ns = not significant (2-way ANOVA, Dunnett’s multiple comparisons test) compared to the negative control. (**b**) It is observed that the smallest areas under the curve (greater anti-inflammatory effect) were shown by G2 IZP 20 mg/kg (AUC = 116.1) and G3 IZP 40 mg/kg (AUC = 116.5) compared to the negative control DMSO 1% (AUC = 641).

**Figure 3 molecules-28-03722-f003:**
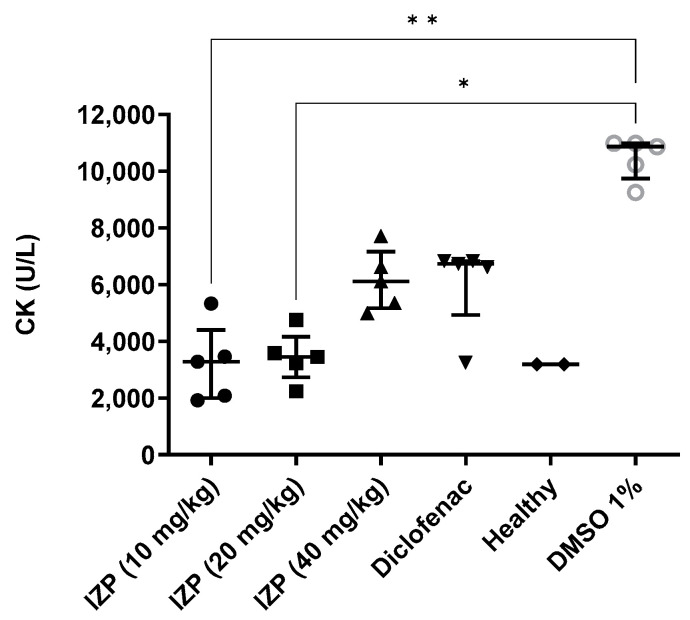
Measurement of serum creatine kinase levels at the end of the experiment, * *p* < 0.05; ** *p* < 0.01; (one-way ANOVA, Dunn’s multiple comparisons test) with respect to the negative control (DMSO 1%).

**Figure 4 molecules-28-03722-f004:**
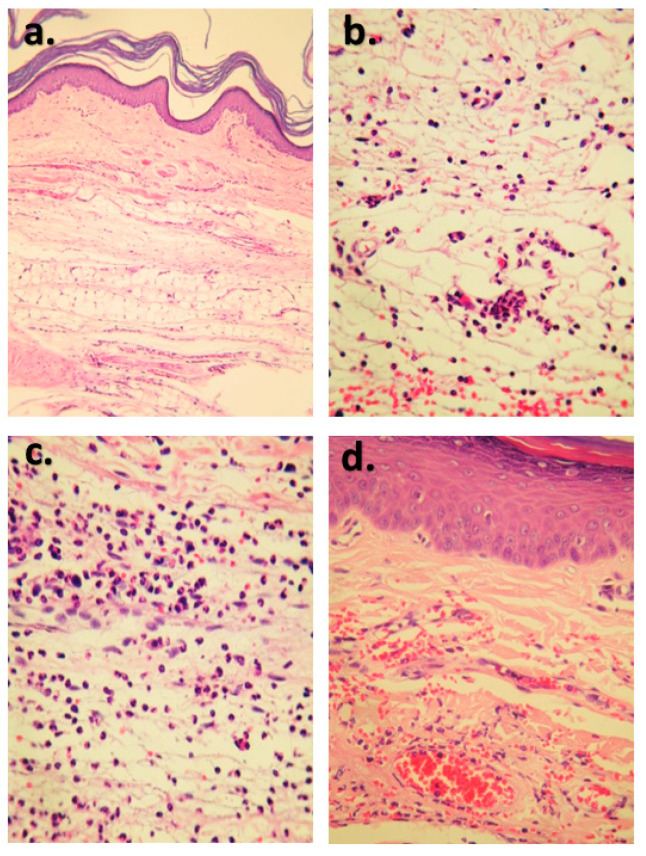
Comparison of histological lesions in rats treated (**b**–**d**) and the control group (healthy animals) (**a**). Note the intensity of the PMN inflammatory cell infiltration in the dermis in the different treatments: (**a**) control group, without inflammatory reaction. H-E. 40×. (**b**) G2 group, treated with 20 mg/kg izalpinine, with moderate PMN infiltration in the dermis. H-E. 40×. (**c**) G4 group treated with 100 mg/kg diclofenac, showing severe PMN infiltration in the dermis. H-E. 40×. (**d**) G5 group treated with 1% DMSO with mild PMN infiltration in the dermis. H-E. 10×. (H-E: Hematoxilin Eosin).

**Figure 5 molecules-28-03722-f005:**
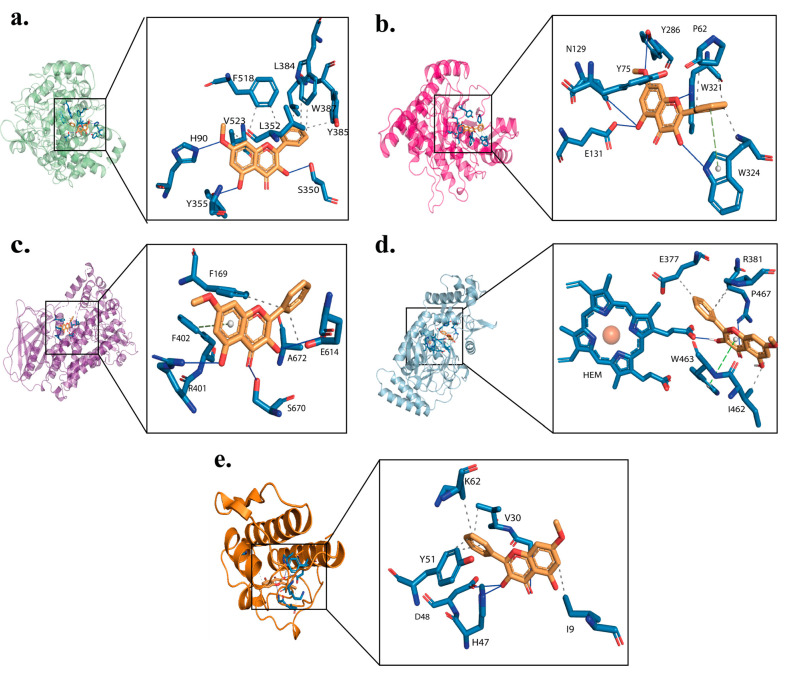
Interaction of IZP with (**a**) COX-2 (A); (**b**) HAase; (**c**) 5-LOX; (**d**) NOS; and (**e**) 5-PLA2. Relevant interactions between the IZP and targets are shown as dashed and continuous lines (blue: H-bond; grey: hydrophobic; green: π–π stacking).

**Figure 6 molecules-28-03722-f006:**
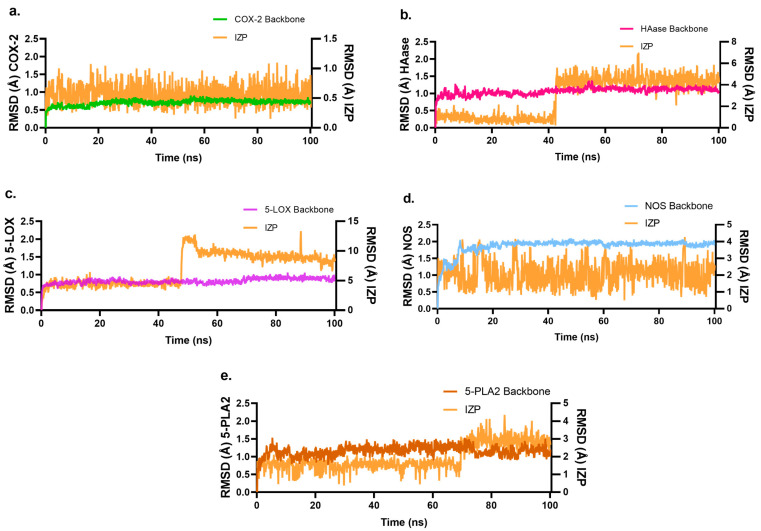
Evolution of RMSD along molecular dynamics for IZP and receptors. (**a**) COX-2; (**b**) HAase; (**c**) 5-LOX; (**d**) NOS; (**e**) 5-PLA2.

**Table 1 molecules-28-03722-t001:** ^1^H NMR (300 MHz) and ^13^ NMR (100 MHZ) spectral data of IZP in DMSO-d_6_.

Position	δ_H_	δc
2	-	146.1
3	9.8(s, OH)	137.4
4	-	176.4
5	12.3 (s, OH)	160.4
6	6.37 (d, *J* = 1.8 Hz, 1H)	97.6
7	-	165.1
8	6.77 (s, *J* = 1.8 Hz, 1H)	92.1
9	-	156.3
10	-	104.2
1′	-	130.8
2′	8.19 (d, *J* = 7 Hz, 1H)	127.5
3′	7.55 (d, *J* = 7 Hz, 2H)	128.5
4′	7.53 (m, *J* = 7 Hz, 2H)	130.0
5′	7.52 (m, *J* = 7 Hz, 2H)	128.5
6′	8.16 (d, *J* = 7 Hz, 1H)	127.5
7-OCH_3_	3.85 (s,3H)	56.07

**Table 2 molecules-28-03722-t002:** Molecular docking results, binding free energies for IZP, and interacting residues.

Target	PDB Code	GlideScore ^a^	ΔG_Bind_ ^b^	Interacting Residues ^c^
COX-2	5KIR	−7.33	−66.07	H90 ^1^, R120 ^6^, Q192 ^1^, Y348 ^3^, V349 ^3^, L352 ^3^, S353 ^1^, Y355 ^3^, F381 ^3^, L384 ^3^, Y385 ^3^, W387 ^3^, R513 ^6^, F518 ^3^, I517 ^3^, A516 ^3^, M522 ^3^, V523 ^3^, A527 ^3^, S530 ^1^
HAase	2PE4	−5.39	−52.14	N37 ^1^, N39 ^1^, P62 ^3^, I73 ^3^, Y75 ^3^, V127 ^3^, E131 ^6^, Y202 ^3^, Y247 ^3^, W321 ^3^, V322 ^3^, S323 ^1^, W324 ^3^
5-LOX	3V99	−4.29	−40.03	S14 ^1^, Q15 ^1^, A18 ^3^, Y81 ^3^, F169 ^3^, A398 ^3^, R401 ^6^, F402 ^3^, I406 ^3^, N554 ^3^, F555 ^3^, F610 ^3^, E614 ^6^, L615 ^3^, E622 ^6^, P668 ^3^, N669 ^1^, S670 ^1^, A672 ^3^.
iNOS	2NSI	−3.18	−32.96	M120 ^3^, I201 ^3^, M374 ^3^, T376 ^1^, E377 ^6^, Arg381 ^6^, W461 ^3^, I462 ^3^, W463 ^3^, V465 ^3^, P466 ^3^, P467 ^3^, HEM550 ^2^
PLA2-IIA	1KQU	−6.89	−57.41	L2 ^3^, F5 ^3,5^, H6 ^1^, I9 ^3^, A17 ^3^, A18 ^3^, Y21 ^3^, H27 ^1^, C28 ^3^, V30 ^3^, C44 ^3^, H47 ^4,5,6^, D48 ^2^, Y51 ^3,5^, K62 ^6^, F98 ^3^.

^a^ GlideScore in kcal/mol. Glide SP visualizer module was used to analyze the docked poses. ^b^ ΔG_Bind_ (kcal/mol) were calculated using the molecular mechanics-generalized born surface area (MM-GBSA) method. ^c^ All interactions were observed within 4Å radius between the ligands and binding site residues. Interaction types: ^1^ polar; ^2^ H-bond; ^3^ hydrophobic; ^4^ π-cation; ^5^ π–π stacking; and ^6^ electrostatic interactions.

**Table 3 molecules-28-03722-t003:** Distribution of groups by treatment.

Groups (G)	Treatment
	Izalpinin (IZP)
1	IZP (10 mg/Kg)
2	IZP (20 mg/Kg)
3	IZP (40 mg/Kg)
4	Positive control (Diclofenac: 100 mg/Kg)
5	Negative control (DMSO 1%)
6	Control group (2) Healthy animals

## Data Availability

Data are contained within the article.

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
