# Peer review of "Anti-Inflammatory Effect of Izalpinin Derived from Chromolaena leivensis: λ-Carrageenan-Induced Paw Edema and In Silico Model"

_molecules, 2023, doi:10.3390/molecules28093722_

Round 1

Reviewer 1 Report

In the manuscript (ID: molecules-2347014), Juan C. Mancipe et al. revealed that izalpinin had anti-inflammatory effect on l-carrageenan-induced paw edema model.

Overall, the manuscript is well-written. However, there are a few suggestions and comments. This will help readers understand better. I recommend considering major and minor revision for publication.

Major points

1. The authors show izalpinin decreases inflammation on l-carrageenan-induced paw edema.

The authors should observe the level of inflammatory and anti-inflammatory cytokine level by ELISA or qPCR.

2. The authors the effect of izalpinin on histological changes in the paw edema have been evaluated by HE staining.

Figure 3 is lacking any quantification to assess how representative these images are.

Quantification of polymorphonuclear cells per area is expected with a graphical presentation of the data and statistical analysis.

3. The authors need to examine total white cell count and the percentage of polymorphonuclear cells or neutrophil in treated group mice blood.

4. The authors show the binding interaction of izalpinin with enzymes.

4-1. The authors need to determine if izalpinin is blocking enzyme activity.

4-2. It would be informative to determine if izalpinin suppress the substance produced by the action of enzymes.

5. In paw edema model, inflammation is suppressed in a concentration-dependent manner of izalpnin, but the opposite result is not obtained with creatine kinase levels.

Minor points.

1. Figure legends should state n numbers for all sample sets.

2. Please use larger fonts to facilitate a better experience for the reviewers and readers in Figure 1.

3. Please include a sescription of the arrows shown in each image in the figure legend.

4. It is difficult to see whether they show some particular patterns at this magnification in Figure 3.

Please show magnified images in Figure 3

5. There is no scale bar in Figure 3.

Please show scale bar. 

Author Response

Major points

  1. The authors show izalpinin decreases inflammation on l-carrageenan-induced paw edema.

The authors should observe the level of inflammatory and anti-inflammatory cytokine level by ELISA or qPCR.

Blood CK levels were quantified by ELISA. Inflammation can cause an increase in creatine kinase (CK) levels in the blood. This is because inflammation can damage cells, including muscle cells, leading to the release of CK into the bloodstream. This is shown in figure 3.

  1. The authors the effect of izalpinin on histological changes in the paw edema have been evaluated by HE staining.

Figure 3 is lacking any quantification to assess how representative these images are.

Quantification of polymorphonuclear cells per area is expected with a graphical presentation of the data and statistical analysis.

Figure 4, which refers to the histopathological study, has been improved. Additionally, the description has been made more detailed to provide clarity to the analysis.

In addition, cell counting was performed using a Zeiss optical light microscope. The PMNs present in five fields were counted using a 100x objective. The counting was performed by two observers and the results were averaged.

The results of the cell counting are shown in a graph and were statistically analyzed. The graph is included in the supplementary material.

  1. The authors need to examine total white cell count and the percentage of polymorphonuclear cells or neutrophil in treated group mice blood.

Cell counting was performed using a Zeiss optical light microscope. The PMNs present in five fields were counted using a 100x objective. The counting was performed by two observers and the results were averaged (Figure S4).

4.The authors show the binding interaction of izalpinin with enzymes.

4-1. The authors need to determine if izalpinin is blocking enzyme activity.

4-2. It would be informative to determine if izalpinin suppress the substance produced by the action of enzymes.

In this section of the research, in vitro assays on enzymes were not performed, but it is planned to do so later in order to establish whether the blockage occurs competitively or non-competitively. Therefore, a computational study was proposed to model how IZP is interacting with enzymes involved in the inflammatory process

  1. In paw edema model, inflammation is suppressed in a concentration-dependent manner of izalpnin, but the opposite result is not obtained with creatine kinase levels.

In Figure 3 (adjusted, significant differences were found between 1% DMSO and the groups treated with IZP at 10 and 20 mg/Kg.), it can be observed that the levels of CK in the treated groups are lower compared to the negative control group (1% DMSO).

Furthermore, it can be seen that the groups treated with doses of 10 and 20 mg/kg exhibit CK levels similar to the healthy control, indicating a reduction in muscle damage compared to the control groups.

Minor points.

  1. Figure legends should state n numbers for all sample sets.

The legends of all figures were adjusted according to the numbering used.

  1. Please use larger fonts to facilitate a better experience for the reviewers and readers in Figure 1.

The font size was increased in Figure 1.

  1. Please include a description of the arrows shown in each image in the figure legend. Figure 3 and its description were improved. 
  2. It is difficult to see whether they show some particular patterns at this magnification in Figure 3.

Please show magnified images in Figure 3

  1. There is no scale bar in Figure 3.

Figure 3 and its description were improved

Reviewer 2 Report

The publication describes a study of the anti-inflammatory effect of isalpine, a compound isolated from Chromolaena leivensis. The article is interesting and the obtained results are significant.

Here are some issues that need to be addressed before an article is accepted:

1.       Line 23: “target proteins involved in the inflammatory process.” Please specify the number or type of these proteins.

2.       The first sentence of the introduction is too long and should be split into two sentences.

3.       1. The manuscript lacks a drawing of the izalpinin structure, which makes it difficult to check the discussion of structural research. It is proposed to add the relevant figure with the designation of A, B and C rings.

4.       Lines 90-92: value 137.4 is not listed in table 1.

5.       What biological tests have been performed previously for isalpinin?

6.       Figure 1 is hard to read when printed in black and white. In the description “positive control” could be replaced with name of used drug.

7.       Line 131: there is no Figure 4 in the manuscript. The description of figure 3 could be more informative.

Author Response

  1. Line 23: “target proteins involved in the inflammatory process.” Please specify the number or type of these proteins

The number of proteins studied was added - Line 24.

  1. The first sentence of the introduction is too long and should be split into two sentences.

The authors consider the introduction to be relevant and appropriate.

  1. The manuscript lacks a drawing of the izalpinin structure, which makes it difficult to check the discussion of structural research. It is proposed to add the relevant figure with the designation of A, B and C rings.

The structure of izalpinin was added - Line 78 and 79

  1. Lines 90-92: value 137.4 is not listed in table 1.

The value in table 1 was corrected

  1. What biological tests have been performed previously for isalpinin?

In the introduction, the studies reported for this compound are referenced, among which are antitumor and analgesic activity - Line 57

  1. Figure 1 is hard to read when printed in black and white. In the description “positive control” could be replaced with name of used drug.

The name 'positive control' was changed to the drug name 'Diclofenac', and the font size of the figure was increased. We hope the journal prints the figure in color

  1. Line 131: there is no Figure 4 in the manuscript. The description of figure 3 could be more informative.

A few lines were added mentioning the Figure - Line 164 and 165.

Round 2

Reviewer 1 Report

The authors answered the reviewer appropriately. I recommend acceptance and publication of this paper in its present form.